# ATase inhibition rescues age-associated proteotoxicity of the secretory pathway

Maeghan Murie[1,2,3,7], Yajing Peng[1,2,7], Michael J. Rigby[1,2,3], Inca A. Dieterich[1,2,3], Mark A. Farrugia[1,2,6], Andreas Endresen[1,2], Anita Bhattacharyya[2,4] & Luigi Puglielli [1,2,5]✉

Malfunction of autophagy contributes to the progression of many chronic age-associated diseases. As such, improving normal proteostatic mechanisms is an active target for biomedical research and a key focal area for aging research. Endoplasmic reticulum (ER)-based acetylation has emerged as a mechanism that ensures proteostasis within the ER by regulating the induction of ER specific autophagy. ER acetylation is ensured by two ER-membrane bound acetyltransferases, ATase1 and ATase2. Here, we show that ATase inhibitors can rescue ongoing disease manifestations associated with the segmental progeria-like phenotype of AT-1 sTg mice. We also describe a pipeline to reliably identify ATase inhibitors with promising druggability properties. Finally, we show that successful ATase inhibitors can rescue the proteopathy of a mouse model of Alzheimer's disease. In conclusion, our study proposes that ATase-targeting approaches might offer a translational pathway for many age-associated proteopathies affecting the ER/secretory pathway.

[1] Department of Medicine, University of Wisconsin-Madison, Madison, WI, USA. [2] Waisman Center, University of Wisconsin-Madison, Madison, WI, USA. [3] Neuroscience Training Program, University of Wisconsin-Madison, Madison, WI, USA. [4] Department of Cell and Regenerative Biology, University of Wisconsin-Madison, Madison, WI, USA. [5] Geriatric Research Education Clinical Center, Veterans Affairs Medical Center, Madison, WI, USA. [6] Present address: Mark A. Farrugia, Michigan State University, East Lansing, MI, USA. [7] These authors contributed equally: Maeghan Murie and Yajing Peng. ✉email: lp1@medicine.wisc.edu

About 75% of all mRNAs are attached to the endoplasmic reticulum (ER) where the bulk of protein biosynthesis occurs[1]. Depending on the cell type, about half of these newly-synthesized proteins will be released into the cytosol while the other half will insert into the ER and engage the secretory pathway[1]. Mechanisms that ensure protein quality control and efficient removal of misfolded/unfolded polypeptides are in place to maintain protein homeostasis (also referred to as proteostasis) in both the cytosol and the ER/secretory pathway[2,3].

ER acetylation has emerged as a mechanism that ensures proteostasis within the ER by regulating the induction of ER specific autophagy (also referred to as reticulophagy or ER-phagy) and the engagement of the secretory pathway[4–11]. ER acetylation is ensured by an ER-membrane transporter, AT-1 (also referred to as SLC33A1), that translocates acetyl-CoA from the cytosol to the ER lumen, and two ER-based acetyl-CoA:lysine acetyl-transferases, ATase1 (also referred to as NAT8B) and ATase2 (also referred to as NAT8), that acetylate ER cargo proteins within the ER lumen[4,12,13]. AT-1/SLC33A1 mutations and gene duplication events that disrupt ER acetylation are linked to severe phenotypes, including developmental delay with childhood death, peripheral neuropathy, and dysmorphism with autism spectrum disorder and intellectual disability[4,14–19].

The removal of toxic protein aggregates through reticulophagy is a particularly important aspect of the proteostatic functions of ER acetylation. Reduced ER acetylation in AT-1 hypomorphic (AT-1$^{S113R/+}$) mice causes increased induction of reticulophagy, while increased ER acetylation in AT-1 overexpressing (AT-1 sTg) mice has the opposite effect[6,8,20]. Knockout of Atase1 (Atase1$^{-/-}$) in the mouse results in activation of reticulophagy, as well as rescue of disease-associated proteotoxicity[11]. Finally, inhibition of ATase1 and ATase2, downstream of AT-1, can restore reticulophagy and resolve disease-associated proteotoxic states[8,9].

Malfunction of autophagy contributes to the progression of many chronic age-associated diseases, including neurodegeneration, cancer, nephropathies, immune and cardiovascular diseases[21–24]. In addition, many chronic degenerative diseases are characterized by the aberrant accumulation of toxic protein aggregates[21–24]. Compelling data indicate that increased levels of autophagy can be beneficial in mouse models of diseases where the accumulation of toxic protein aggregates represents a major pathogenic component[25–29]. As such, improving normal proteostatic mechanisms is an active target for biomedical research and a key focal area for aging research[21–24]. A major limitation for autophagy-based translational approaches is the ability to selectively target autophagy to a specific cellular location, and the ability to rapidly test prospective autophagy-stimulating compounds in a relevant mouse model.

ATase1 and ATase2 are both type-II membrane proteins; they have a very short cytosolic tail, a membrane spanning domain, and a catalytic domain facing the lumen of the ER[4,13]. The ER-bound autophagy protein 9A (ATG9A) is the immediate target for the ATase-mediated regulation of reticulophagy[4,7–9,30]. The acetylation of ATG9A regulates its ability to interact with FAM134B and SEC62, two ER-based autophagy receptors, and engage LC3β and the core autophagy machinery in the cytosol[4,8,10,11]. Importantly, inhibition of ATase1/ATase2 stimulates reticulophagy and helps dispose of toxic protein aggregates that form in the ER lumen/secretory pathway but not in the cytosol[9]. Therefore, translational approaches targeting the ATases would be congenial to rescue proteopathies that are specifically caused by proteotoxicity within the ER/secretory pathway.

Here, we describe a pipeline to reliably identify ATase inhibitors with promising druggability properties. We also show that these ATase inhibitors can rescue disease manifestations associated with the progeria-like phenotype of AT-1 sTg mice, as well as the proteopathy of the Alzheimer's disease (AD)-like phenotype of APP/PS1 mice. Therefore, our study suggests that ATase-targeting approaches might offer a translational pathway for several age-associated proteopathies affecting the ER/secretory pathway.

## Results

**Late treatment with Compound 9 rescues the AT-1 sTg phenotype.** We previously reported that early treatment of AT-1 sTg mice with the ATase1/ATase2 inhibitor, Compound 9, was able to prevent the proteostatic defects as well as the progeria-like phenotype of the animals[8]. A limitation of that study is that treatment began at weaning, when some of the disease phenotypes were not severe. Therefore, to assess whether ATase inhibition can also rescue severe and ongoing disease manifestations, we started treatment of AT-1 sTg mice at 2 months of age, a point where they already display a severe phenotype (Fig. 1a).

Compound 9 was administered orally (50 mg kg$^{-1}$ day$^{-1}$) in the form of food pellets[8,9]. Within two months, the animals looked completely normal and indistinguishable from their WT littermates (Fig. 1a). Treatment rescued their lifespan (Fig. 1b) as well as their ability to gain weight (Fig. 1c). Post-mortem assessment revealed that Compound 9 was able to rescue the systemic inflammation, as reflected by the lymphadenopathy (Fig. 1d) and the tissue IgG infiltration (Fig. 1e); as well as the accumulation of senescent cells within peripheral organs, as reflected by senescent-associated β-Galactosidase (SA-β-Gal) activity (Fig. 1f), p21 mRNA levels (Fig. 1g), and p16 protein levels (Fig. 1h). Compound 9 also protected AT-1 sTg mice from all other major disease manifestations that characterize their progeria-like phenotype, such as body fat loss (Fig. 2a), loss of bone density (Fig. 2b), splenomegaly (Fig. 2c), and anemia (Fig. 2d). Finally, treatment was able to restore the Atg9a-Fam134b and Atg9a-Sec62 interaction on the ER (Fig. 2e–g), which is essential for the proteostatic regulatory functions of the ER acetylation machinery[8,10,11].

When taken together, the above results indicate that inhibition of both ER-based ATases through Compound 9 does not only prevent the development or the exacerbation of disease phenotypes, as demonstrated in our previous study[8]; it also rescues ongoing severe disease manifestations of the AT-1 sTg progeria-like phenotype. Therefore, they expand our previous findings and suggest translational potential for a variety of age-associated diseases with clinically detectable manifestations.

**Levels of human AT-1 increase with age.** We previously reported that AT-1 levels increase in primary mouse neurons as a function of their age in culture. We also reported that AT-1 levels are higher in the brain of p44$^{+/+}$ mice, an established progeria-like model[12]. To determine whether AT-1 levels increase in humans as a function of age, we determined the mRNA levels of AT-1 in primary human skin fibroblasts. We observed a significant and progressive age-dependent increase with no evident plateau (Supplementary Fig. 1a), suggesting that the results obtained with AT-1 sTg mice can -at least in part- inform us on proteotoxic states associated with diseases of age. Interestingly, we did not detect an increase of either ATase1 or ATase2 mRNA levels (Supplementary Fig. 1b, c). To complement these data, we also analyzed AT-1 mRNA levels in the brain of the NIA aging mouse C57BL/6 cohort but did not observe significant changes (Supplementary Fig. 1d; discussed later).

**ATase inhibitors can be identified through in silico binding.** We recently reported that ATase1 and ATase2 are differentially

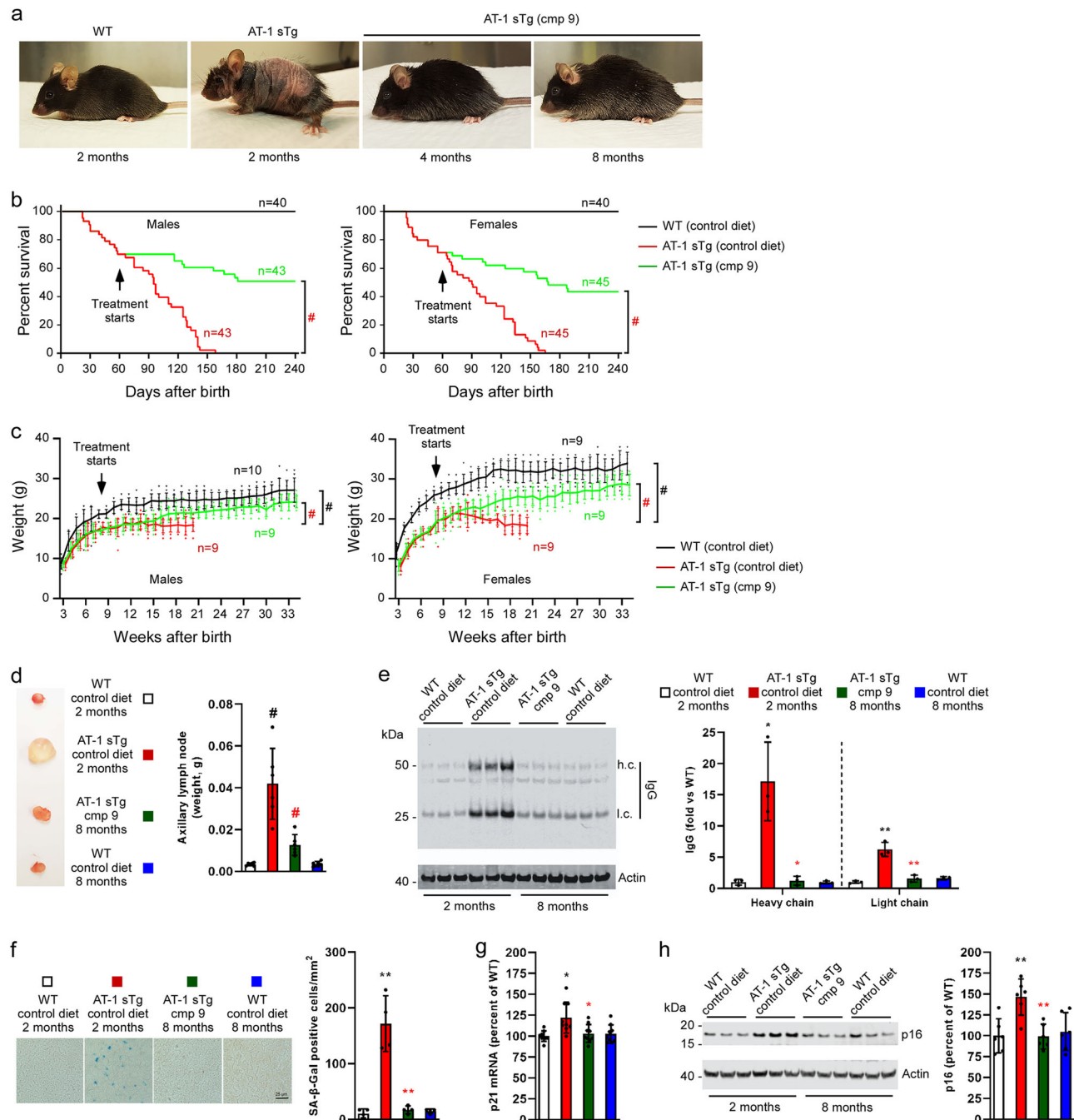

**Fig. 1 Late treatment with Compound 9 rescues the AT-1 sTg phenotype. a** Representative images of WT mice at 2 months of age and AT-1 sTg mice at 2, 4, and 8 months of age. Compound 9 treatment started at 2 months of age. **b** Lifespan of AT-1 sTg mice on control diet versus Compound 9 diet. **c** Body weight of AT-1 sTg mice on control diet versus Compound 9 diet. **d** Representative image (left panel) and weight quantification (right panel) of axillary lymph nodes. **e** Representative Western blot (left panel) and quantification (right panel) of IgG heavy chain (h.c.) and light chain (l.c.) levels in liver. **f** Representative images (left panel) and quantification (right panel) of SA-β-Gal staining in liver. **g** p21 mRNA quantitation in liver. **h** Representative Western blot (left panel) and quantification (right panel) of p16 levels in liver. Bars represent mean ± SD. *$p < 0.05$, **$p < 0.005$, #$p < 0.0005$ via ordinary two-way ANOVA with Tukey's multiple comparison test. Black symbol, significance vs WT; Red symbol, significance vs untreated AT-1 sTg.

regulated[31]. Importantly, ATase1 has an allosteric switch that can link its acetyltransferase activity to the influx of acetyl-CoA into the ER lumen while ATase2 does not[31]. Therefore, ATase1 is likely the primary target in those situations where increased AT-1 levels and activity are at the basis of the phenotype. This would include patients with *AT-1/SLC33A1* duplications as well as normal human aging where increased expression of AT-1 might be disrupting normal proteostasis within the ER and secretory pathway. By studying Atase1$^{-/-}$ and Atase2$^{-/-}$ mice, we also

discovered that these two acetyltransferases have partially divergent functions with ATase1 playing a more fundamental role in the regulation of reticulophagy[11]. Therefore, compounds that primarily target ATase1 rather than ATase2 are predicted to have stronger translational potential for disease states associated with dysfunctional ER proteostasis.

Compound 9 was initially identified using a high throughput screen of a library of small molecules that yielded 30 ATase inhibitors[32]. To further understand the mechanism of action of

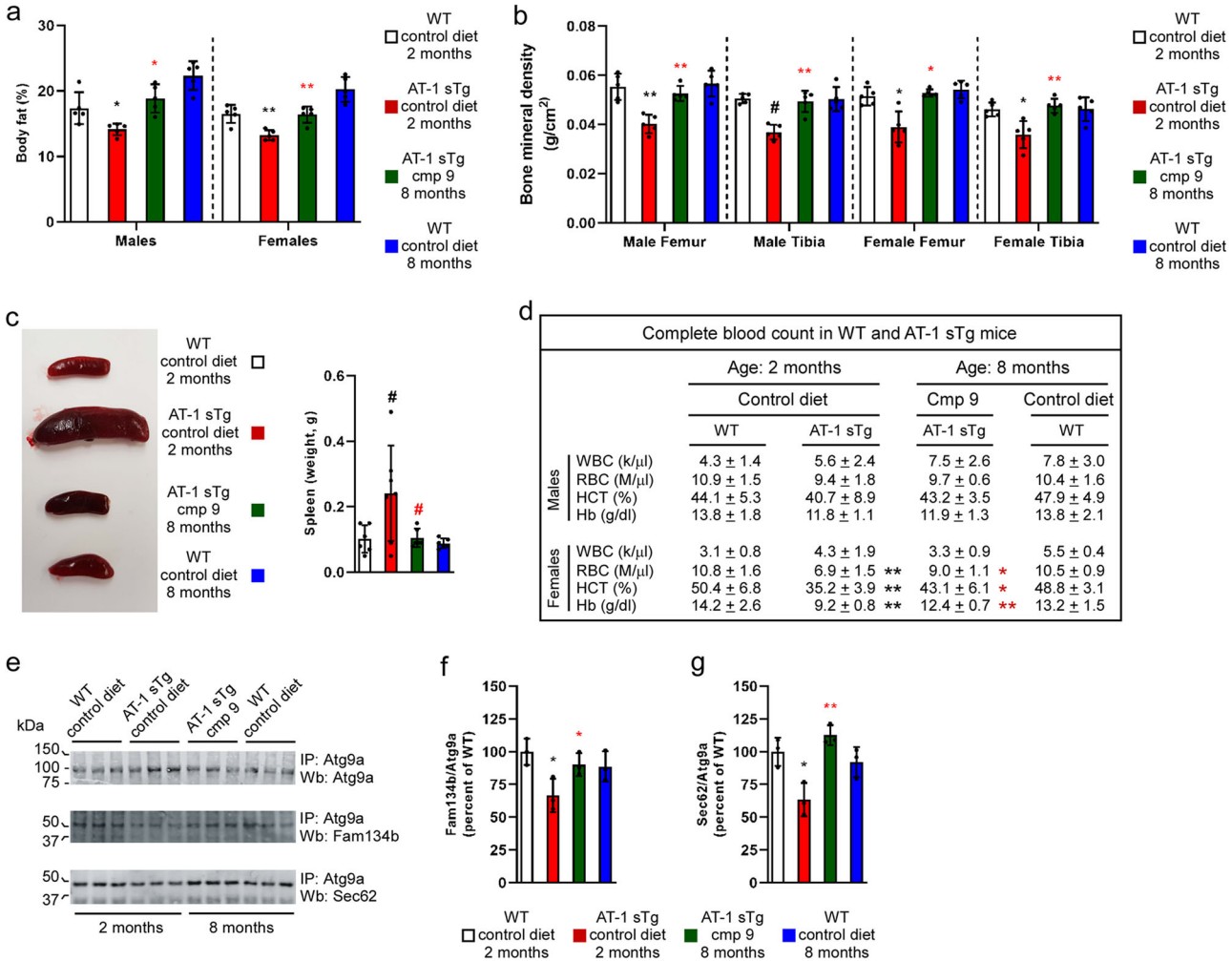

**Fig. 2 Late treatment with Compound 9 rescues the AT-1 sTg phenotype. a** Quantification of body fat. **b** Quantification of bone mineral density.
**c** Representative image of spleens (left panel) and quantification (right panel). **d** Hematologic parameters of WT mice and AT-1 sTg mice (WT (2 month),
$n = 5$; WT (8 month), $n = 5$; AT-1 sTg (control diet), $n = 5$; AT-1 sTg (cmp 9), $n = 5$). **e–g** Western blot showing co-immunoprecipitation of Atg9a,
Fam134b, and Sec62 in WT and AT-1 sTg mice. Representative blots are shown in (**e**) while quantitation of results is shown in (**f, g**). Bars represent mean ±
SD. *$p < 0.05$, **$p < 0.005$, #$p < 0.0005$ via ordinary two-way ANOVA with Tukey's multiple comparison test. Black symbol, significance vs WT; Red
symbol, significance vs untreated/control diet AT-1 sTg. WBC white blood cells; RBC red blood cells; HCT hematocrit; Hb hemoglobin.

the above inhibitors and improve our ability to select additional
molecules, all 30 compounds underwent optimization for *in silico*
docking and were then assessed for potential binding affinities for
both ATase1 and ATase2. Docking for ATase1 revealed that the
compounds segregated in two groups inserting with different
binding energy either in the acetyl-CoA pocket or in the predicted
peptidyl-Lys pocket (Fig. 3a, b). Docking for ATase2 yielded
slightly different results with compounds mostly clustering at the
interspace between the two pockets (Fig. 3c, d).

Through binding affinity assessment and molecular profiling
(Lipinski's rule), we selected three additional compounds for
further in vivo analysis, Compound 10, Compound 11, and
Compound 19. They all displayed binding energy values that were
in the range of acetyl-CoA (Fig. 3b, d). Individual docking of the
compounds are shown in Fig. 3e, f. Variability in where the
molecules were predicted to bind is present within the binding
affinity placements for each compound (Fig. 3e–g). Particularly,
Compound 11 had poor binding affinity in most of the predicted
placements for ATase1 (Fig. 3g). The inhibitory properties of the
compounds determined by the initial high throughput screen
were as follows: *Cmp 9*, 74% ATase1, 88% ATase2; *Cmp 10*, 52%
ATase1, 80% ATase2; *Cmp 11*, 43% ATase1, 74% ATase2; *Cmp*

19, 56% ATase1, 74% ATase2. The inhibitory properties of all
compounds listed in Fig. 3b, d are reported in Supplementary
Table 1. The chemical structure of the four compounds is shown
in Fig. 3h. Importantly, they all follow the criteria set by a
modified Lipinski's Rule as being potential central nervous system
(CNS) drugs[33–35] (Fig. 3i). The Lipinski calculations for all
compounds listed in Fig. 3b, d are reported in Supplementary
Data 1.

**ATase inhibitors rescue the AT-1 sTg phenotype.** To assess the
effectiveness of the newly identified ATase inhibitors, we admi-
nistered the compounds to AT-1 sTg mice following the same
protocol and dose design that we used before (see Figs. 1, 2 and
refs. [8,9]). Specifically, Compounds 10, 11, and 19 were adminis-
tered orally (50 mg kg$^{-1}$ day$^{-1}$) in the form of food pellets.
Treatment began at weaning.

All three compounds were able to rescue the AT-1 sTg
phenotype. Indeed, we observed a drastic rescue of the general
appearance of the animals (Fig. 4a) and their lifespan (Fig. 4b).
Postmortem assessment showed that treatment was able to
prevent all major phenotypic features of AT-1 sTg mice, namely
loss of body fat (Fig. 4c), loss of bone mineral density (Fig. 4d),

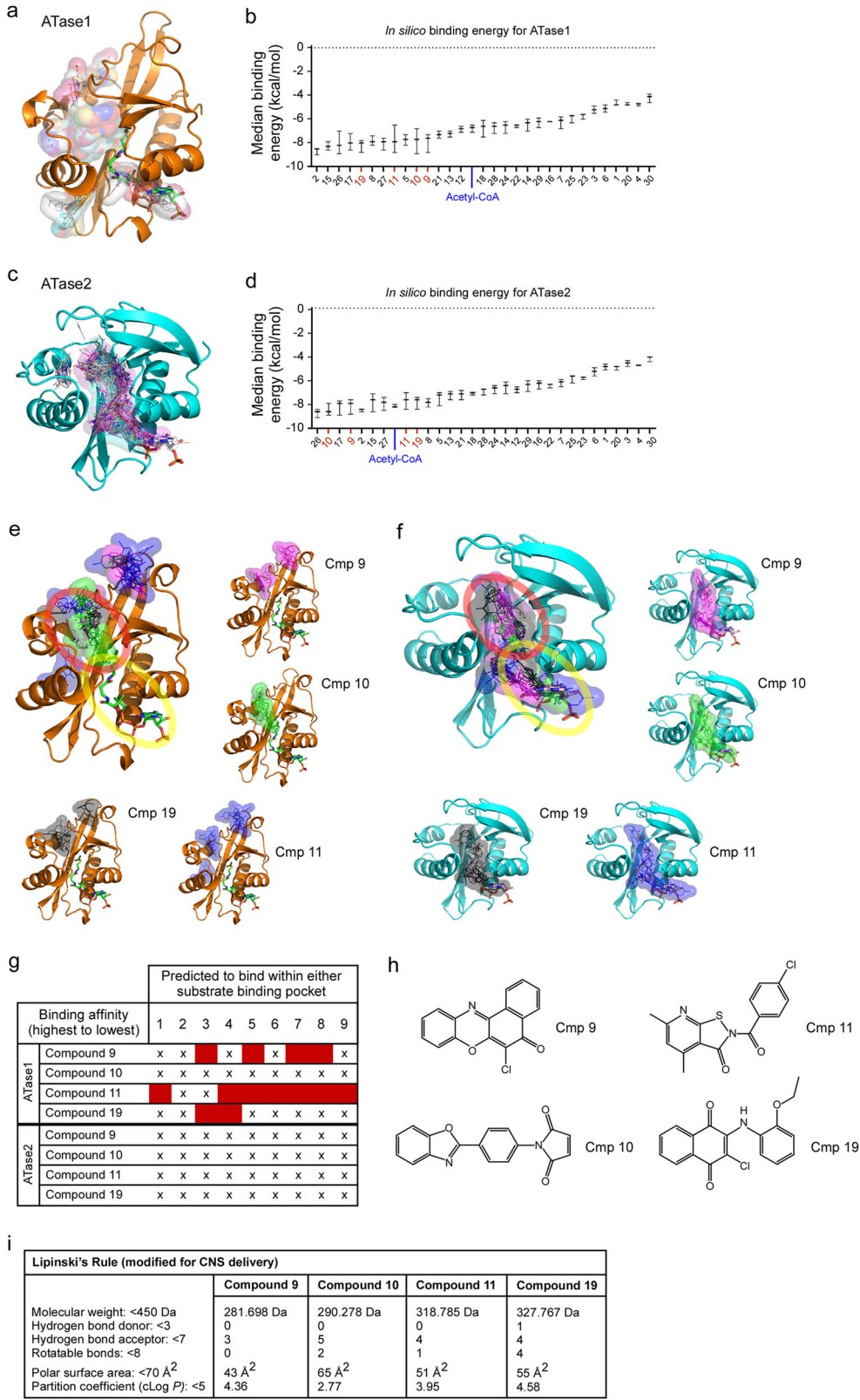

**Fig. 3 Identification of ATase inhibitors through in silico binding.** Representative images (**a** and **c**) and median binding energy (**b** and **d**) of potential ATase inhibitors that are docked *in silico* to ATase1 (**a**, **b**) or ATase2 (**c**, **d**). Representative images of Compound 9, 10, 11, and 19 bound *in silico* to ATase1 (**e**) or ATase2 (**f**). The top 8 binding affinity placements are shown for each compound. The peptidyl-Lys binding pocket is circled in red while the acetyl-CoA binding pocket is circled in yellow. **g** Prediction of binding within substrate pockets for ATase1 and ATase2. Top 8 binding affinity placements are listed for each compound. Binding affinity placement within either the peptidyl-Lys or the acetyl-CoA pocket is marked with an (X); binding affinity placement outside of either pocket is marked by a red box. **h** Structures of Compound 9, 10, 11, and 19. **i** Modified Lipinski's Rule for CNS delivery of Compound 9, 10, 11, and 19.

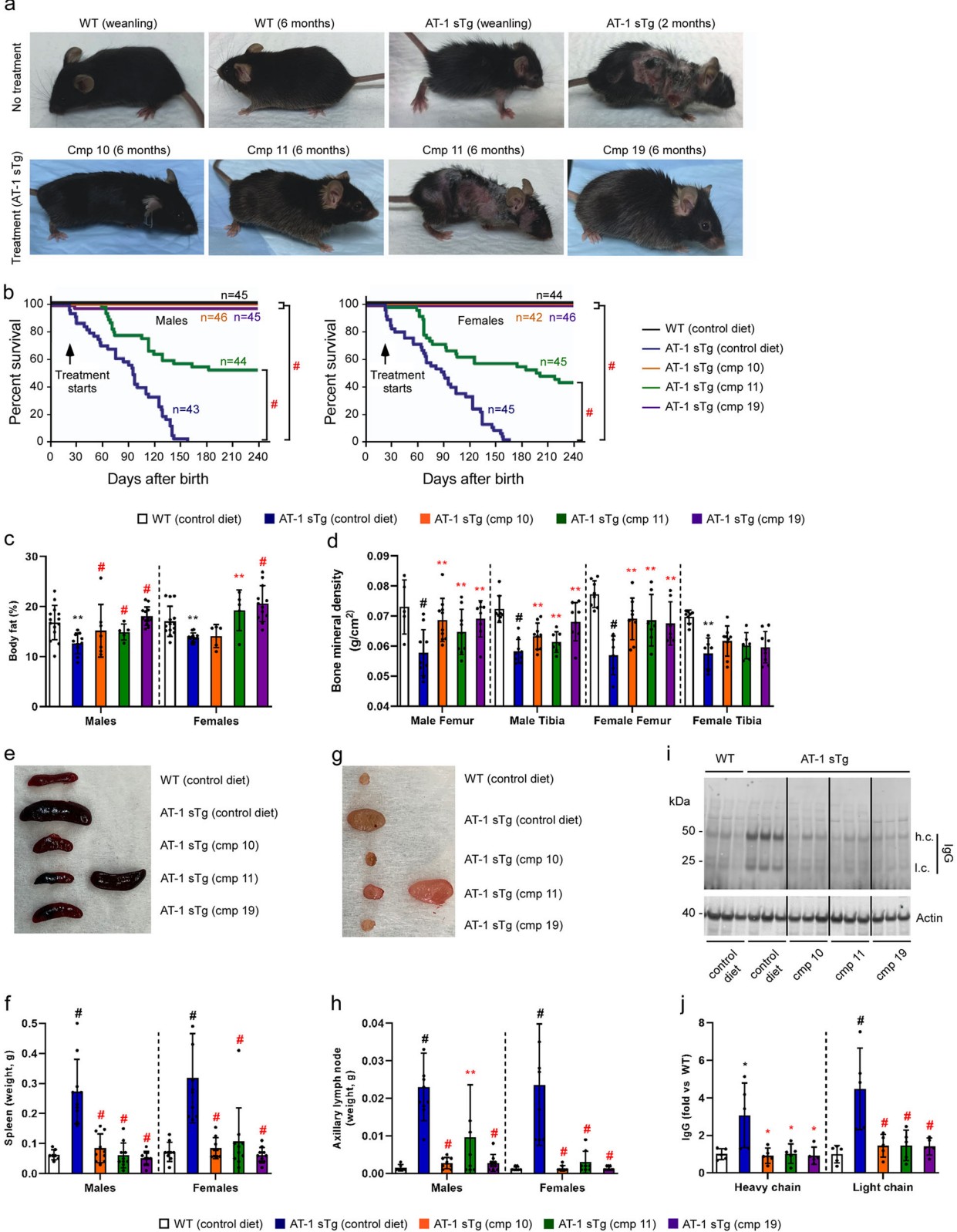

**Fig. 4 ATase inhibitors rescue the AT-1 sTg phenotype. a** Representative images of WT and AT-1 sTg mice with and without treatment. **b** Lifespan of WT mice on control diet and AT-1 sTg mice on either control or Compound 10, 11, or 19 diets. Treatment started at weaning. **c** Quantification of body fat. **d** Quantification of bone mineral density. **e** Representative image of spleens. **f** Quantification of (**e**). **g** Representative image of axillary lymph nodes. **h** Quantification of (**g**). **i** Representative Western blot showing IgG heavy chain (h.c.) and light chain (l.c.) levels in liver. **j** Quantification of (**i**). For **c–j** mice were studied at approximately 3 months of age. Bars represent mean ± *SD*. *$p < 0.05$, **$p < 0.005$, ***$p < 0.0005$, #$p < 0.0001$ via ordinary two-way ANOVA with Tukey's multiple comparison test. Black symbol, significance vs WT; Red symbol, significance vs untreated/control diet AT-1 sTg.

splenomegaly (Fig. 4e, f), lymphadenopathy (Fig. 4g, h), and tissue inflammation (Fig. 4i, j). An important outcome of the above results is that Compound 10 and 19, which displayed high binding affinity for ATase1 (Fig. 3g), generally showed complete phenotypic rescue, while Compound 11, which displayed poor binding affinity to ATase1 (Fig. 3g), showed variability in the degree of rescue across all parameters (Fig. 4).

**ATase inhibitors reduce Alzheimer's disease pathology in APP/PS1 mice.** AD is one of the most common age-associated diseases, and a well-characterized form of age-associated proteopathy. We previously reported that the levels of AT-1, as well as both ATase1 and ATase2, are higher in late-onset AD patients as compared to age-matched controls[12,32]. We also reported that reduced ER acetylation, as observed in AT-1 hypomorphic (AT-1$^{S113R/+}$) mice or following ATase1/ATase2 inhibition by Compound 9, can resolve the proteopathy associated with the AD-like phenotype of APP$_{695/swe}$ mice[9].

To evaluate the therapeutic potential of the ATase inhibitors, Compounds 10, 11, and 19 were administered to APP/PS1 mice. Again, all compounds were given to the animals at weaning and were administered orally (50 mg kg$^{-1}$ day$^{-1}$) in the form of food pellets. Mouse brains were assessed at 10 months of age, when key features of AD-like pathology are well apparent. By using 6E10-based immunohistochemistry, we observed reduced Aβ plaque area coverage in both the cortex and hippocampus of APP/PS1 mice treated with Compound 10 and 19 (Fig. 5a, b). Similar findings were observed with Thioflavin-S staining of dense plaques (Fig. 5c, d). To determine whether the reduced plaque pathology was accompanied by parallel rescue of the synaptic loss that typically characterizes the APP/PS1 phenotype, we stained for the pre-synaptic protein synaptophysin and the post-synaptic protein Psd-95. Both Compound 10 and 19 were able to rescue the synaptic loss as demonstrated by the increased number of co-localized puncta compared to the untreated mice (Fig. 5e, f). Additionally, we evaluated the phosphorylation status of tau by Western blotting for pTau pSer396 and found that Compound 19 was able to reduce tau phosphorylation (Fig. 5g, h). As with AT-1 sTg mice, Compound 11 had limited effect on the progression of the APP/PS1 phenotype (Fig. 5).

**Compound 9 and 19 treatment decreases the acetylation status of Atg9a.** We previously reported that the acetylation status of ER-bound ATG9A regulates the induction of reticulophagy downstream of the ATases[7-11,30]. To determine whether the levels of acetylated ATG9A can be useful to monitor target engagement and predict rescue of proteostatic dysfunctions in vivo, we treated AT-1 sTg mice with either Compound 9 or 19 and studied the timeline of the progeria-like phenotype as a function of ATG9A acetylation.

First, we evaluated the pharmacokinetic parameters of both compounds (Supplementary Table 2), and then we evaluated the levels of acetylated-Atg9a in AT-1 sTg mice treated with either compound. The acetylation status of ER-bound Atg9a was compared to the timeline of disease progression. As before, treatment began at the age of 2 months, a point when disease features were already evident. Compound 9-treated mice manifested some improvement after one week of treatment, although the phenotypic changes were clearly evident at the second week (Fig. 6a). The timeline of lysine acetylation of ER-bound Atg9a appeared to mirror the same progression (Fig. 6b, c). Similar results were observed following Compound 19 treatment (Fig. 6d–f). Again, the reduced acetylation of ER-bound Atg9a matched the rescue of the disease manifestations.

## Discussion

The ER acetylation machinery has emerged as a mechanism that ensures proteostasis within the ER and secretory pathway by regulating the induction of reticulophagy and the engagement of the secretory pathway[5-11]. The removal of toxic protein aggregates through reticulophagy is a particularly important aspect of the proteostatic functions of ER acetylation. Reduced ER acetylation in AT-1 hypomorphic (AT-1$^{S113R/+}$) mice causes increased induction of reticulophagy, while increased ER acetylation in AT-1 overexpressing (AT-1 sTg) mice has the opposite effect[6,8,20]. Knockout of Atase1 (Atase1$^{-/-}$) in the mouse results in activation of reticulophagy, and knockout of either Atase1 or Atase2 results in activation of macroautophagy[11]. Finally, biochemical inhibition of ATase1 and ATase2, downstream of AT-1, can restore reticulophagy in AT-1 overexpressors, and genetic disruption of the Atases can rescue features of the AD-associated proteotoxicity[8,9,11] (see also this study). The results presented in this study indicate that ATase inhibition not only prevents the progression of the progeria-like phenotype of AT-1 sTg mice, but also rescues ongoing disease manifestations of the animals, thus suggesting that inhibition of the ATases and recovery of proteostasis within the ER might have translational potential for diseases characterized by defective ER proteostasis. These include patients with duplications of AT-1/SLC33A1 as well as mutations that affect folding of proteins that insert within the ER/secretory pathway causing them to misfold and aggregate[36-40]. Defective ER proteostasis has also been implicated in the pathogenesis and/or progression of different age-associated degenerative diseases, including AD[4,9,21,24,29]. Consistently, in addition to the progeria-like phenotype of AT-1 sTg mice, our study indicates that inhibition of ER acetylation can rescue several features of the AD-like phenotype of APP/PS1 mice. Importantly, similar results were also observed in APP$_{695/swe}$ mice indicating that the effect of ATase inhibition is not limited to just one mouse model of AD[9].

We previously reported that AT-1 levels increase in primary mouse neurons as a function of their age in culture; we also reported that AT-1 levels are higher in the brain of p44$^{+/+}$ mice, an established progeria-like model[12]. Now, we report that AT-1 levels increase in normal human skin fibroblasts as a function of age. These findings suggest that our studies with AT-1 sTg mice can -at least in part- inform us on proteotoxic states associated with diseases of age. In this study, we did not find changes in AT-1 mRNA levels in the brain of the NIA aging C57BL/6 mouse cohort. The divergent response of human skin fibroblasts and mouse brain might reflect inherent experimental differences including the complexity of the brain tissue and the metabolic status of cells in culture versus brain tissue in vivo. Alternatively, it may also reflect differences in the transcriptional regulation of AT-1 in the two species. Importantly, AT-1 levels were found to be upregulated in the brain of late-onset AD patients as compared to age-matched controls thus expanding the potential impact of our studies to one of the most common forms of age-associated dementias[12,41].

Interestingly, we did not observe upregulation of either ATase in our human fibroblast study. However, we previously reported upregulation of both ATases in the brain of late-onset AD patients[32]. Therefore, it is possible that humans aging primarily affects ER acetylation through AT-1 while AD affects both AT-1 and the ATases. It is also possible that the transcriptional regulation of the ATases is differentially regulated in different cell types. Indeed, we have recently reported the existence of functional REST/NRSF binding sites, which serve to repress neuronal genes in non-neuronal cells, within the coding region of both genes[31]. It is also worth stressing that ATase1 has an allosteric switch while ATase2 does not[31]. The allosteric switch would allow ATase1 to immediately respond to the levels of acetyl-CoA within

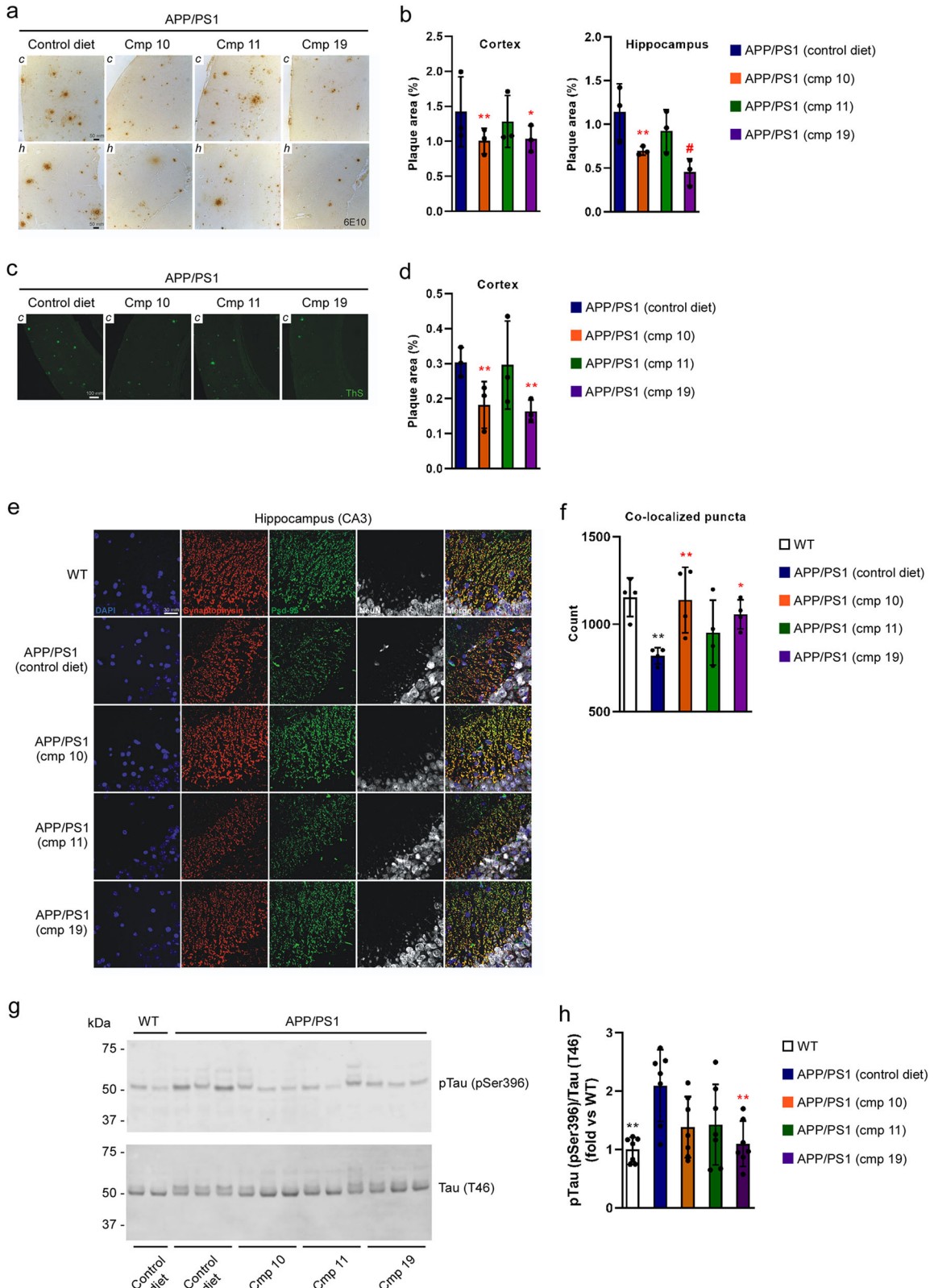

the ER lumen, thus coupling acetyltransferase activity to the rate of acetyl-CoA transport by AT-1. This is similar to the allosteric regulation of histone acetylation by acetyltransferase p300[42]. The above findings should be interpreted together with our recent observation that ATase1 and ATase2 have slightly different biological functions with ATase1 playing a primary role with the regulation of reticulophagy[11,31]. In essence, ATase1 appears to be the primary translational target with diseases characterized by defective ER proteostasis.

In light of the fact that the vast majority of potential therapeutic compounds fail in clinical trials, developing a large repertoire of potential inhibitors is essential. We previously reported the identification and characterization of Compound 9 as the first ATase-specific inhibitor[32]. Here, we used *in silico*

**Fig. 5 ATase inhibitors Compound 10 and Compound 19 reduce Alzheimer's disease pathology in the APP/PS1 mouse. a** 6E10 immunohistochemistry in 5 μm paraffin-embedded brain slices. **b** Quantification of (**a**). **c** Thioflavin-S staining of 5 μm paraffin-embedded cortex brain slices. **d** Quantification of (**c**). **e** Immunostaining of 5 μm paraffin-embedded brain slices for the presynaptic marker synaptophysin and postsynaptic marker Psd-95. NeuN (neuronal marker) is included for anatomic reference. **f** Quantification of (**e**). Red and green puncta of 1 μm diameter were fit with spots and those within 1 μm of one another were counted. **g** Representative Western blot showing levels of pTau (pSer396) in the brain. **h** Quantification of (**g**). Mice were 10-months old at their end-point. Bars represent mean ± SD. *$p < 0.05$, **$p < 0.005$, #$p < 0.0005$ via ordinary two-way ANOVA with Dunnett's multiple comparison test (for 6E10 immunohistochemistry and Thioflavin-S staining) or with Tukey's multiple comparison test (for immunostaining of synaptic markers and Western blotting for Tau phosphorylation). In **b**, **d**, and **f**, each data point represents the average of technical replicates (a single brain slice) for a given animal. Black symbol, significance vs WT; Red symbol, significance vs untreated/control diet AT-1 sTg.

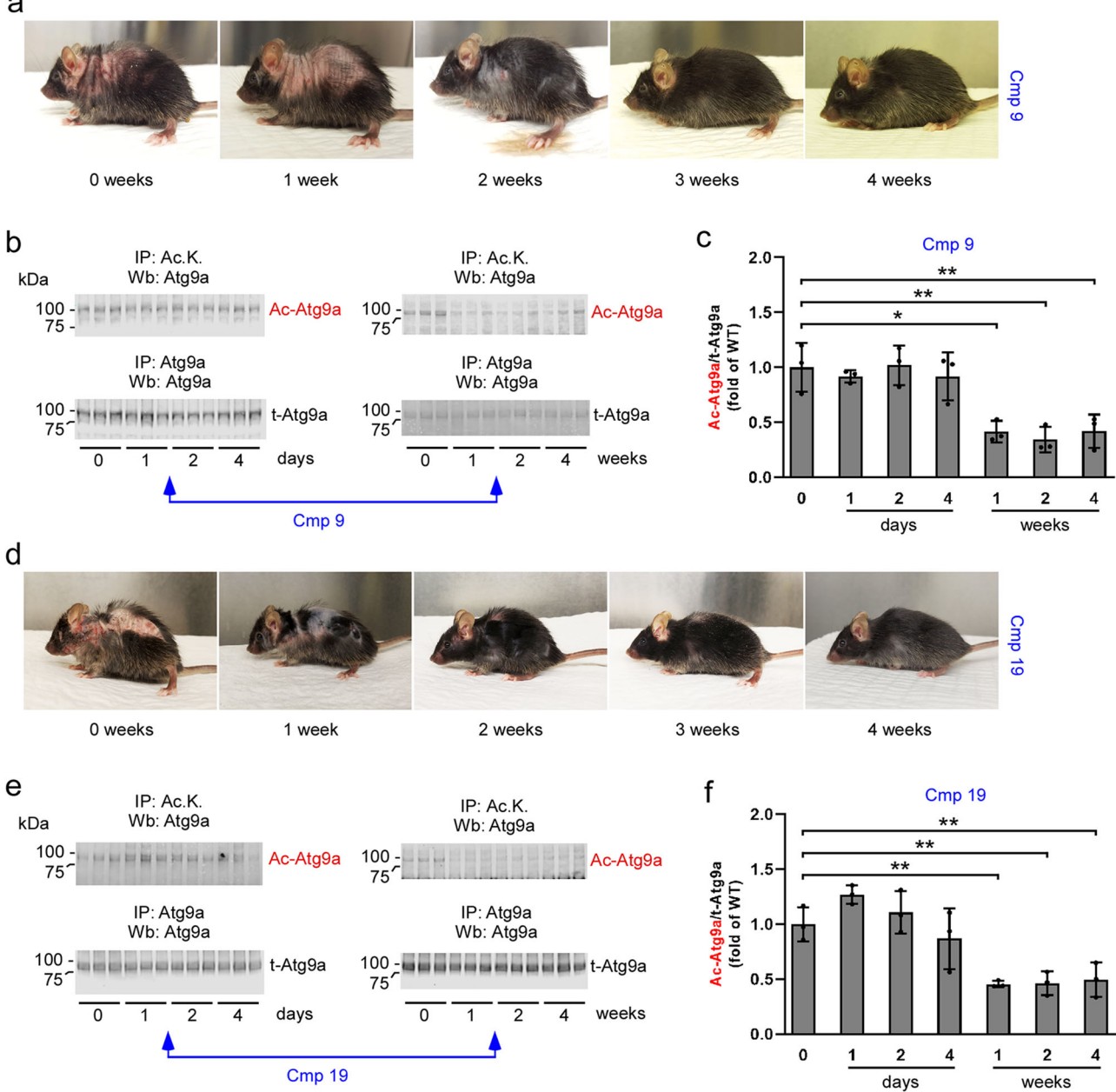

**Fig. 6 Target engagement of Compound 9 and Compound 19 in AT-1 sTg mice. a** Representative images to document rescue of the phenotype following Compound 9 treatment. The same AT-1 sTg mouse is shown. **b** and **c** Western blot showing levels of acetylated-Atg9a (Ac-Atg9a) in ER preparations from liver. Representative blots are shown in (**b**) while quantitation of results is shown in (**c**). **d** Representative images to document rescue of the phenotype following Compound 19 treatment. The same AT-1 sTg mouse is shown. **e** and **f** Western blot showing levels of acetylated-Atg9a (Ac-Atg9a) in ER preparations from liver. Representative blots are shown in (**e**) while quantitation of results is shown in (**f**). Bars represent mean ± SD. *$p < 0.05$, **$p < 0.005$, ***$p < 0.0005$, #$p < 0.0001$ via ordinary two-way ANOVA plaque area percentage with Tukey's multiple comparison test. Treatment started at 2-months of age.

docking and molecular profiling to optimize our screening approach and selected three additional ATase inhibitors. Two of them, Compound 10 and Compound 19, displayed strong affinities for both ATase1 and ATase2, while one, Compound 11, displayed strong affinity only for ATase2. Docking also revealed quite different inhibitor modalities of our Compounds. Indeed, in the case of ATase1, most compounds displayed docking within the predicted peptidyl-Lys pocket, while, in the case of ATase2, most of the compounds displayed docking at the interface between the two substrate-binding pockets. These differences likely reflect the slightly different structural features of the two acetyltransferases, and might inform us on future translational studies.

Interestingly, Compound 10 and 19, which displayed high binding affinities for both ATases displayed stronger translational effects with both the AT-1 sTg and the APP/PS1 mouse models, while Compound 11, which displayed high binding affinity only for ATase2, had limited effects, thus supporting the argument of ATase1 being the primary translational target. However, we also must be aware of different pharmacokinetic/pharmacodynamic (PK/PD) properties that might affect biological outcomes. Here, we only performed an initial and limited PK/PD study of Compound 9 and 19. A more comprehensive assessment should be envisioned to compare different ATase inhibitors prior to possible optimization. Although with limitations, the *in silico* docking approach that we used appears to provide a reliable initial screening tool to identify ATase inhibitors that may be useful as therapeutics.

Mechanistically, the regulation of reticulophagy involves ATase-mediated acetylation of the autophagy protein ATG9A within the lumen of the ER[8–11,30]. The acetylation status of ATG9A -in turn- regulates the ability of ATG9A to interact with FAM134B and SEC62, and engage cytosolic LC3β, thus activating reticulophagy[8,10]. Therefore, the acetylation status of ATG9A can be used to determine the successful target engagement of prospective ATase inhibitors. Our present studies further strengthen the above conclusions as changes in ATG9A acetylation immediately preceded the phenotypic improvement of AT-1 sTg mice.

In conclusion, our study demonstrates that inhibition of the ATases restores proteostatic functions within the ER and can rescue the disease phenotype of AT-1 sTg, a mouse model of segmental progeria, and APP/PS1, a mouse model of AD. Our study also supports the case that ATase inhibitors might have translational potential to resolve proteotoxic states that characterize age-associated diseases. Finally, we describe a pipeline to reliably identify ATase inhibitors with promising druggability properties. Major components of this pipeline are the *in silico* docking and the availability of a mouse model of segmental progeria with defective ER proteostasis. The *in silico* docking can help select the most promising compounds that emerge from a classical high throughput screen, while AT-1 sTg mice can ensure rapid in vivo testing of potential inhibitors. Finally, we demonstrate that the acetylation status of ER-bound ATG9A can serve to monitor target engagement and predict in vivo rescue of proteostatic dysfunction.

## Methods

**Animals**. AT-1 sTg mice were generated by crossing Rosa26:tTA mice with pTRE-AT-1 mice[43]. Genotyping from tail DNA was performed at weaning in house or by Transnetyx (Cordova, TN) using the following primers: AT-1 forward (5′-AAT CTG GGA AAC TGG CCT TCT-3′), AT-1 reverse (5′-TAT TAC CGC CTT TGA GTG AGC TGA-3′), Rosa forward (5′-AAA GTC GCT CTG AGT TGT TAT-3′), and Rosa reverse (5′-GCG AAG AGT TTG TCC TCA ACC-3′). Both males and females were studied with wild-type (WT) littermates used as controls. AT-1 sTg mice were studied at different ages, as appropriately reflected in the specific figures and legends.

APP[695/swe]/PS1-dE9 (APP/PS1) double transgenic mice were obtained from Jackson Laboratory (MMRRC Stock No. 34832-JAX). Genotyping from tail DNA was performed at weaning by Transnetyx (Cordova, TN) using the following primers: APP[SW] forward (5′-CCG ACA TGA CTC AGG ATA TGA AGT T-3′) and APP[SWE] reverse (5′-CCT TTG TTT GAA CCC ACA TCT TCT G-3′). Male mice were studied with WT littermates used as controls. APP/PS1 mice were studied at the age of 10 months.

All mice were housed in standard cages provided by the University Laboratory Animal Resources and grouped with littermates with 1–5 mice per cage. Animals were provided water *ad libitum* and supplied either standard chow or a compound fortified diet. The rodent diet with Compound 9, 10, 11, or 19 was manufactured by Bio-Serv.

Brain tissue from the aging C57BL/6 mouse line was provided by the NIA Aged Rodent Tissue Bank: https://www.nia.nih.gov/research/dab/aged-rodent-tissue-bank-handbook”).

**Human skin fibroblasts**. Human fibroblasts were obtained from the University of Wisconsin-Madison Human Stem Cell Core and from Coriell (Supplementary Table 3) with approval from the University of Wisconsin Human Subject IRB. All individuals were considered healthy, at the time of collection. Fibroblasts were maintained in TFM media (DMEM, 15% tetracycline-free FBS, 1× NEAA) with media changes every other day. Cells were grown to 80% confluency on a six-well plate before collection.

**Real-time PCR analysis of AT-1, ATase1, and ATase2**. Real-time PCR (qPCR) was performed using the Roche 480 lightcycler and Sybr Green Real Time PCR Master Mix (04707416001, Roche). Target gene expression levels were normalized against *GAPDH* levels and expressed as raw target to reference values ($2^{-\Delta\Delta Ct}$). PCR primers specific to each gene are: human *AT-1/SLC33A1* forward 5′-CAGGCGG TTGGGATGACAGT-3′ and reverse 5′-AAGATTTGCGACGACCGAAGTT-3′; human *ATase1/NAT8B* forward 5′-GGCCAGTCCTTCTTCCAC-3′ and reverse 5′-ATAGACGCGCCTG CCTGAGC-3′; human *ATase2/NAT8* forward 5′-GGCCAGTC CTTCTTCTGT-3′ and reverse 5′-TCACAGACTCCCTACCTTAGA-3′; human *GAPDH* forward 5′-TTTGTCAAGCTCATTTCCTGGTA-3′ and reverse 5′-TTCAA GGGGTCTACATGGCAACTG-3′; mouse *Slc33a1* forward 5′-TACGTGCTT-CAGGGCATTCC-3′ and reverse 5′-CTGAAGAAAGCCTGGTCTGTATAGC-3′; mouse *Gapdh* forward 5′-GTTGTCTCCTGCGACTTCA-3′ and reverse 5′ GGTG GTCCAGGGTTTCTTA-3′. The cycling parameters were as follows: 95 °C for 10 s; 58 °C (human primers), or 55 °C (mouse primers) for 20 s; and 72 °C for 30 s; 45 cycles.

**ER isolation**. The total ER isolation was prepared using a commercial ER Enrichment kit (Novus Biologicals), according to the manufacturer's protocol. Briefly, 0.5 g of liver tissue was homogenized in isosmotic homogenization buffer using a Dounce Teflon homogenizer for 20–30 strokes. The homogenized tissue was centrifuged at 1000 × *g* for 10 min at 4 °C; the supernatant was centrifuged again at 12,000 × *g* for 15 min at 4 °C, discarding the pellet afterwards. Finally, the supernatant was centrifuged at 90,000 × *g* for 60 min at 4 °C to obtain total ER fraction as a pellet. The pellet was resuspended in the provided suspension buffer and subjected to Western blot analysis or immunoprecipitation.

**Protein extraction**. Protein extracts were prepared in GTIP buffer (10 mM Tris, pH 7.6, 2 mM EDTA, 0.15 M NaCl) supplemented with 1% TritonTM X-100 (Roche Applied Science), 0.25% Nonidet P-40 (Roche Applied Science), complete protein inhibitor mixture (Roche Applied Science), and phosphatase inhibitors (mixture set I and set II; Calbiochem)[43,44] The tissue was homogenized by using a Dounce Teflon homogenizer for 20 strokes followed by sonication on an ethanol ice bath for 1 ×30 s with a 1 min interval between each sonication. The homogenized mixture was then centrifuged at 4000 × *g* for 10 min at 4 °C. The supernatant was taken for experimentation and the pellet was discarded.

**Western blotting and immunoprecipitation**. Protein concentration was measured by the bicinchoninic acid method (Pierce) and protein electrophoresis was performed on a NuPAGE® system using 4–12% Bis-Tris gels (Invitrogen). The following primary antibodies were used: anti-p16 (1:1000; Abcam; ab189034), anti-ATG9A (1:1000; Abcam, clone EPR2450(2); ab108338), anti-FAM134B (1:4000; Abcam; ab151755), anti-Sec62 (1:1000; Abcam; ab140644), anti-pTau pSer396 (1:1000; Cell Signaling; 9632S), anti-Tau T46 (1:2000; ThermoFisher; 13-6400), and anti-β-actin (1:1000; Cell Signaling Technology; 4967). Anti-p16, anti-ATG9A, and anti-β-actin blots were visualized with goat anti-rabbit or anti-mouse Alexa Fluor® 680-conjugated or Alexa Fluor® 800-conjugated secondary antibodies on infrared imaging (LICOR Odyssey Infrared Imaging System; LI-COR Biosciences). To visualize anti-Fam134b or anti-Sec62, mouse anti-rabbit TrueBlot HRP-conjugated secondary antibody (Rockland; #18-8816-31) was used followed by chemiluminescent detection with Amersham ECL Western Blotting Detection Kit (GE Healthcare; #GERPN2209) on the Azure c600 imager (Azure Biosystems). For enriched liver ER Western blotting, target proteins were normalized to total protein staining performed before immunodetection (LiCor; #926-11010).

Immunoprecipitation was performed on protein extracts (300 μg) from total ER fractions[8,9,30]. Immunoprecipitation was performed using anti-acetylated lysine (1:50; Cell Signaling Technology; 9441) or anti-ATG9A (1:50; Abcam, clone EPR2450(2); ab108338) antibodies and rotated overnight at 4 °C. Approximately 15 μl of washed BioMag protein A magnetic particles (Polysciences, Inc.) were added per sample and rotated an additional 2–3 h at 4 °C. The BioMag protein A magnetic particles were then suspended with a magnet, washed 3× with PBS, resuspended with 1× loading buffer, and denatured at 95 °C for 5 min. The magnetic particles were then resuspended, and the supernatant was loaded onto gels for Western blot analysis.

The original uncropped Western blot images included in the manuscript can be found in Supplementary Figs. 2 and 3.

**Senescent-associated P21 real-time PCR**. Real-time PCR for *P21* was performed using the following cycling parameters: 95 °C, 10 s; 55 °C, 10 s; 72 °C, 15 s, for a maximum of 45 cycles[8,12]. Gene expression levels were normalized against GAPDH levels and expressed as percent of control. Controls without reverse 5 transcription were included in each assay. Specific primers used are: P21 forward (5′-GTG ATT GCG ATG CGC TCA TG-3′), p21 reverse (5′-TCT CTT GCA GAA GAC CAA TC-3′), GAPDH forward (5′-AGG TCG GTG TGA ACG GAT TTG-3′), and GAPDH reverse (5′-TGT AGA CCA TGT AGT TGA GGT CA-3′).

**Senescence-associated β-galactosidase staining**. Cryosections of mouse liver were stained with Senescence β Galactosidase Staining kit (Cell Signaling Technology) according to the manufacturer's protocol and as previously described[8]. Briefly, mouse liver cryosections (10 μm) were fixed with fixation solution for 15 min. Fixed sections were then stained with β-galactosidase at 37 °C overnight in a dry incubator without carbon dioxide. The percentage of senescent cells was expressed as the total number of stained senescent cells divided by the total number of cells counted using immunofluorescence.

**Faxitron radiography and dual-energy X-ray absorptiometry (DEXA)**. Bones were fixed in 70% ethanol and soft tissues were removed from the bone once fixed. Bone mineral density (BMD) and total body fat mass were determined using the UltraFocus DXA system (Faxitron) or a Hewlett Packard Faxitron X-ray system (24 KV for 1.3 min, model 43855A; Hewlett Packard, McMinnville, OR) following standard manufacturer protocols. Calibrations were performed with a phantom of known density, and quality assurance measurements were performed prior to BMD measurements.

**Whole blood analytes**. Blood was collected transcardially from mice with an insulin syringe and collected in BD Microtainer® tubes with K₂EDTA, as previously described[8]. Hematologic parameters were measured on a HemaVet complete blood count (CBC) instrument.

**Histology and immunostaining**. For histology and immunohistochemistry, tissues were collected immediately after euthanasia, fixed overnight, and paraffin-embedded using standard techniques[6,8,9,45]. Thioflavin-S staining was conducted by incubating deparaffinized and rehydrated slides for 10 min in 1% thioflavin-S (Sigma-Aldrich; #T1892-25G) dissolved in 50% ethanol. Slides were rinsed in 80% ethanol and 50% ethanol for 1 min each, briefly rinsed in distilled water, and mounted with aqueous mounting media with DAPI (Electron Microscopy Sciences; #17985-50). The following primary antibodies were used: anti-Beta Amyloid (clone 6E10, 1:100, Signet), anti-NeuN (EMD Millipore; #ABN91MI; 1:1000), anti-synaptophysin (abcam; #ab32127; 1:200), and anti-PSD95 (Thermo Fisher; #MA1-045; 1:200).

Bright-field images were acquired using a Leica DM4000 B microscope with a 10× or 20× air objective using Image-Pro version 6.3. For thioflavin-S stained slides, single z-slice images (1024 × 1024 pixels; 1.24 μm/pixel) were acquired using a 10× air objective (NA = 0.3) at a pinhole size of 129.0 μm. For synaptophysin/Psd-95/NeuN-stained slides, z-stack images (1024 × 1024 pixels at 0.21 μm/pixel with 15 z-stacks at 0.2 μm step size) were acquired on a 60× oil immersion objective (NA = 1.4) at a pinhole size of 39.6 μm.

ImageJ version 2.0 was used in plaque analysis on 6E10 and thioflavin-S images by making binary images via an intensity threshold and counting objects using the particle analyzer. Imaris version 9.5 (Bitplane) was used to quantify synaptic loss by creating 1 μm-diameter red and green spots and counting the number of co-localized spots within 1 μm of each other[11].

**In silico docking**. In silico docking was performed using the software AutoDock Vina version 1.1.2[46]. Optimized models for ATase1 and ATase2 were prepared as previously described[31]. Briefly, 3D structures of the compounds of interest were obtained from the ZINC15 database. These structures had the appropriate polar hydrogens place and were then docked to the optimized models for ATase1 and ATase2 using a search grid covering the entire volume of the protein space with exhaustiveness setting 8 in AutoDock Vina 1.1.2. Binding energetics were output as

mean binding free energy and all orientations were visualized using Python Molecular Viewer (PMV) version 1.5.6 and PyMOL version 2.4.1 softwares[47,48].

**Pharmacokinetics**. Pharmacokinetic studies were conducted by Pharmaron. Briefly, male C57BL/6 WT mice were given 10 mg/kg of vehicle suspended Compound 9 or 19 *per os*. Blood was collected in plastic micro centrifuge tubes containing K₂EDTA via dorsal metatarsal vein or cardiac puncture at 0.5, 1, 2, 4, and 8 h from administration. Whole blood was then centrifuged at $4000 \times g$ for 5 min at 4 °C to obtain plasma. Concentrations of Compounds 9 or 19 in the plasma samples were analyzed using LC-MS/MS and the pharmacokinetic calculations were conducted using WinNonlin (PhoenixTM, version 8.2).

**Statistics and reproducibility**. Data analysis was performed using GraphPad Prism version 8.4.3. Comparison of the means was performed using Student's *t*-test for two groups and ordinary one-way or two-way ANOVA for ≥3 groups followed by either Tukey–Kramer (comparison between all groups) or Dunnett's (comparison to one control group) multiple comparisons test. Data are expressed as mean ± SD unless otherwise specified. Outliers were removed using Grubb's test. Differences were declared statistically significant if $p < 0.05$, and the following statistical significance indicators are used: $*p < 0.05$; $**p < 0.005$; $\#p < 0.0005$.

**Reporting summary**. Further information on research design is available in the Nature Research Reporting Summary linked to this article.

## Data availability

The authors declare that all other data supporting the findings of this study are available within the paper and Supplementary Data (see also Supplementary Data 2).

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

## Acknowledgements

This research was supported by NIH (NS094154, AG053937, and AG057408), the Department of Veterans Affairs (I01 BX004202), and a core grant to the Waisman Center from NIH/NICHD-U54 HD090256. M.M. was supported by the Louis Stokes Alliance for Minority Participation-Bridge to the Doctorate (NSF HRD-1612530) and the NIH (T32 AG000213). M.J.R. was a recipient of the Wisconsin Distinguished Graduate Fellowship.

## Author contributions

M.M., Y.P., M.J.R., I.A.D., M.A.F. and A.E. performed the experiments and analyzed the data. A.B. provided reagents. A.B. and L.P. provided critical advice for the experiments. L.P. designed the overall study. M.M., Y.P. and L.P. wrote the manuscript with input from all authors.

## Competing interests

The authors declare no competing interests.

## Ethics

All animal experiments were carried out in accordance with the National Institute of Health Guide for the Care and Use of Laboratory Animals and were approved by the Institutional Animal Care and Use Committee of the University of Wisconsin-Madison (protocol #M005120). Human fibroblasts were obtained from the University of Wisconsin-Madison Human Stem Cell core and from Coriell with approval from the University of Wisconsin Human Subject IRB (protocol # 2014-0613).
