## [Peer Review File · Communications Biology]

Reviewers' comments:

Reviewer #1 (Remarks to the Author):

Protein toxicity is a major contributor of many diseases including those that affect the brain. However, the place in which protein toxicity occurs may be different from disease to disease, thus requiring carefully targeted therapeutics. ER is among the most affected location wherein protein misfolding and toxicity occur, partially because a large portion of the cellular proteome undergoes translation, folding, and degradation (proteostasis) in the ER. In this study, Murie et al. show that inhibiting ATases that localize in the ER can rescue ongoing disease manifestations of AT-1 sTg mice. Furthermore, the authors identified other inhibitors of ATases, which were shown to have a therapeutic value in both AD and AT-1 sTg mice.

Overall the quality of the manuscript is great and the results from the experiments seem compelling. However, the manuscript lacks significant novelty. Two major novel findings this manuscript advances are: 1) the rescue of AT-1 sTg mice can occur even at a later stage by Compound 9 and 2) other inhibitors found in this study (e.g. Compound 10 and 19) can also mitigate disease progression in both AD and AT-1 sTg mice. Although these are significant findings, previous studies have already shown the efficacy of ATase inhibitor in rescuing disease phenotypes in both AD and AT-1 sTg mice (Peng et al., Brain 2016; Peng et al., Aging Cell 2018). The differences between this study and the previous studies are the Compounds used and time point at which intervention was made. Furthermore, mechanisms underlying the rescue remain unclear in this study. Therefore, this reviewer would like for the authors to adequately address the following comments as a way to improve/clarify the manuscript for publication in Communications Biology.

Major comments:

1. The last part of the abstract (lines 31-37) is exactly the same as the last part of the introduction (lines 96-103). Rephrasing is needed.
2. An important control is missing in some of the figures. Have the authors tested the effect of Compound 9 (and later on 10, 11, and 19) on control mice?
3. Figure 3, although interesting, does seem a little out of place. If it is not essential for the main conclusion of the paper, this figure should be moved to the supplementary figure. On the other hand, further investigating the mechanism by which AT-1 levels increase with age may add to the novelty to this paper. The findings from this could further be applied to the AD mice (e.g. is it the same mechanism in AD? Also, can knocking down or inhibiting AT-1 up-regulator in AD mice rescue phenotypes?)
4. In lines 109-110, the authors stated that the limitation of a previous study was that the "treatment began at weaning, before disease manifestation was fully evident." Thus, in this study, the authors began treatment after the disease manifestation, at the age of 2 months. However, when testing the efficacy of Compounds 10, 11, and 19, the authors reverted to the method of treatment at weaning. What is the logic behind this? What kind of effect do Compounds 10, 11, and 19 have on the disease manifestation of AT-1 sTg mice when treated at 2 months of age?
5. Furthermore, when testing Compounds 10, 11, and 19, the authors did not choose to compare (or show?) their efficacy with that of Compound 9. If the authors have the data, they should show them. If not, it would be helpful to explain why such comparison wasn't made directly. With current data, it is difficult to tell which Compound (9, 10, or 19) is most effective and to be pursued further as a potential therapeutic agent.
6. Although it is implied throughout the manuscript, how ATase inhibitors such as Compounds 10 and 19 rescue disease phenotypes remain unclear. Is it through activation of reticulophagy? If the authors believe that it is, can the authors design a study (perhaps in a cell culture if mice studies take too long) where a rescue by ATase inhibitors are stymied by autophagy (or reticulophagy-specific) inhibitors? Or perhaps another reticulophagy-inducing drugs other than ATase inhibitors (if there are any) might be used to mimic rescue phenotypes of AT-1 sTg mice. With current data, it is difficult to say how ATase inhibitors are rescuing the disease phenotype.

Minor comments:

1. In previous paper (Peng et al., Aging Cell 2018), the authors stated that Compound 9 was administered at weaning when "the initial disease phenotypes were already manifested." This seems to contradict the claim made in this paper (lines 128-130), which states that the previous paper showed that Compound 9 prevented the development of disease phenotypes. This portion should be revised or clarified.
2. Lines 170-171 state that acetyl-CoA is one of the natural ATase substrates. What are other natural substrates of ATase and does Compound 9 block binding of ATase to those substrates as well?
3. Lines 176-177 are a little unclear. Expand upon how the commonalities and differences can illuminate their biochemical properties.
4. By what reasoning did the authors chose to further explore Compounds 10, 11, and 19? The basis for choosing those three seems to be on binding affinity assessment (line 168), but the results of that for the other compounds are not shown. It would be helpful to see the results for the rest of the compounds in the supplementary material.
5. Line 177. Cite Lipinski's Rule.
6. Supplementary Figures should be properly labeled.

Reviewer #2 (Remarks to the Author):

This manuscript outlines an expanded set of lead compounds related to the previously described C9 with improved specificity for ATase1 over ATase2. Interestingly, C9 is shown to modify disease progression post-symptom onset (it was previously only investigated in a prevention model).

Overall the experiments are appropriately powered and the data is well presented. I only have a small number of queries/concerns that should be addressed to improve the strength of the manuscript prior to publication.

- 1) Is there any difference in the housing/handling of the AT-1 sg control animals compared to the treatment groups in Fig.5b prior to the onset of treatment? Particularly in the females it appears there is divergence in survival prior to commencing dosing with the drugs.
- 2) Several of the the lead compounds, particularly C9 share many features with previously described neuronal autophagy-stimulating compounds (Svetkov et al., PNAS 2010 PMID:20833817). Do the most potent compounds in this series (10-NCP and trifluoperazine) show in silico binding activity for ATase 1/2 and if so how dothey compare to C9/C19?
- 3) Please provide survival data for the entire cohort used in Fig.7 to show whether C19's increased specificity for ATase1 is therapeutically beneficial. It is difficult to assess whether the rescued mice shown in the images are representative of the group or hyperresponding outliers.

Reviewer #3 (Remarks to the Author):

This is an excellent work within a well written manuscript. The implications of this work is significant given the number of people affected by ER stress/autophagy disease. Testing in AD mice is an excellent addition and shows other common conditions may benefit from this treatment approach.

The only experiment that I would suggest that would improve the impact of the work is testing levels of AT-1 in young and aged mice to complement the in vitro fibroblast analysis.

All other comments are minor.

Fig 1b the red dots should be moved in front of green so that both can be observed by reader.

Line 186 where the authors state all 3 compounds rescue phenotype does not agree with the information presented in fig with compound 11. Compound 11 doesn't appear to rescue - perhaps authors should say initial observations suggest all 3 rescue but with evaluation of additional animals compound 11 has variable rescue level. This is stated later but would be more consistent with fig if clarified early in paragraph.

Fig 2 please define which 2 experimental groups are being compared for statically significance. Presumably the compound treated mice are compared to untreated and show statistically significant difference. Might be nice to compare compound treated mice to controls to see if there is any difference in these groups.

POINT-BY-POINT RESPONSE

We wish to thank the Editor the Reviewers for their positive comments and suggestions. A comprehensive point-by-point response can be found below.

Summary of changes

We performed all additional experiments and included all necessary revisions. Specifically:

- (i) We clarified the use of compound 9 in control mice.
- (ii) We analyzed AT-1 levels in the mouse brain as a function of age. For this component, we imported tissue from the NIA aging C57BL/6 mouse cohort.
- (iii) We included the efficacy of the four compounds.
- (iv) We clarified the mouse handling.
- (v) We clarified the *in silico* binding activity of compound 9 and 19.

To accommodate new data we added three Tables (Table 1-3) to the Supplementary Material section. All changes within the manuscript are highlighted.

Reviewer #1

Major comments:

1. *The last part of the abstract (lines 31-37) is exactly the same as the last part of the introduction (lines 96-103). Rephrasing is needed.*

Response: Done as requested.

2. *An important control is missing in some of the figures. Have the authors tested the effect of Compound 9 (and later on 10, 11, and 19) on control mice?*

Response: This comment is unclear. If the Reviewer is referring to lifespan-based studies in WT mice, they are outside of our interest. The purpose of this paper is to determine the effect of ATase inhibitors on disease manifestations associated with a progeria-like (AT-1 sTg mice) and an AD-like (APP/PS1 mice) phenotype.

We used WT mice (i) for pre-formulation and formulation development, (ii) to assess immediate potential toxic effects, and (iii) to determine some of the PK properties. (i) and (ii) were performed before initiating the actual treatment with our disease models. The PK parameters of Compound 9 and 19 are shown in Supplementary Table 3.

3. *Figure 3, although interesting, does seem a little out of place. If it is not essential for the main conclusion of the paper, this figure should be moved to the supplementary figure.*

Response: Done as requested.

On the other hand, further investigating the mechanism by which AT-1 levels increase with age may add to the novelty to this paper. The findings from this could further be applied to the AD mice (e.g. is it the same mechanism in AD?)

Response: Some initial studies on the mechanisms that regulate the expression of the entire ER acetylation machinery (AT-1, ATase1 and ATase2) are published in *J Neurochem* 2020;154: 404-423. PMCID: PMC7363514.

Also, can knocking down or inhibiting AT-1 up-regulator in AD mice rescue phenotypes?

Response: Yes. Haploinsufficiency of AT-1 (AT-1^{S113R/+} mice) rescues the AD-like phenotype of APP mice (see *Brain* 2016; 139: 937-952. PMCID: PMC4805081). Genetic disruption of ATase1 (Atase1^{-/-} mice) or ATase2 (Atase2^{-/-} mice) rescues the AD-like phenotype of APP/PS1 mice (see *Commun Biol* 2021; 4: 454. PMCID: PMC8041774). The studies performed with ATase1 and ATase2 knock-out mice also confirmed that ATase1 is a better target than ATase2. These studies are mentioned in the current manuscript.

4. In lines 109-110, the authors stated that the limitation of a previous study was that the “treatment began at weaning, before disease manifestation was fully evident.” Thus, in this study, the authors began treatment after the disease manifestation, at the age of 2 months. However, when testing the efficacy of Compounds 10, 11, and 19, the authors reverted to the method of treatment at weaning. What is the logic behind this? What kind of effect do Compounds 10, 11, and 19 have on the disease manifestation of AT-1 sTg mice when treated at 2 months of age?

Response: The “weaning treatment” is usually the approach of choice for translational studies employing novel compounds in the mouse. This explains our strategy in the *Brain* 2016 paper with APP mice, the *Aging Cell* 2018 paper with AT-1 sTg mice, as well as the cmp 10/11/19 studies included in the second part of the current manuscript.

The “2-month treatment” used with AT-1 sTg mice in the first part of this study goes beyond the “usual” strategies and has the purpose of examining whether on-going severe disease manifestations can be rescued. Conceptually, these results prove the point and, therefore, we felt that there was no need to repeat the same study with the other compounds. Furthermore, the main purpose of the second part of the paper is to determine whether we can streamline *in silico* hits into a mouse model where therapeutic potential can be evaluated in a very short period (2 to 4 months). As such, the “weaning approach” was more logical. Follow-up “late studies” with more compounds can certainly be performed. However, they are beyond the scope of this manuscript.

5. Furthermore, when testing Compounds 10, 11, and 19, the authors did not choose to compare (or show?) their efficacy with that of Compound 9. If the authors have the data, they should show them. If not, it would be helpful to explain why such comparison wasn't made directly. With current data, it is difficult to tell which Compound (9, 10, or 19) is most effective and to be pursued further as a potential therapeutic agent.

Response: Inhibition properties determined by the initial HTS have been introduced in the text. The main point of the figure is to show that the *in silico* analysis can help predict – at least in part – *in vivo* efficacy. However, we do agree with the Reviewer that these numbers can be informative. The entire dataset is now included in the Supplementary Material.

6. *Although it is implied throughout the manuscript, how ATase inhibitors such as Compounds 10 and 19 rescue disease phenotypes remain unclear. Is it through activation of reticulophagy? If the authors believe that it is, can the authors design a study (perhaps in a cell culture if mice studies take too long) where a rescue by ATase inhibitors are stymied by autophagy (or reticulophagy-specific) inhibitors?*

Response: The ER acetylation machinery maintains proteostasis within the ER and secretory pathway. A recent review on the topic can be found in *J Cell Sci* 2018; 131: jcs.221747. PMID: PMC6262770. Mechanistically, the regulation of reticulophagy is achieved through the acetylation status of the autophagy protein 9A (ATG9A) within the ER lumen, which -in turn- regulates the interaction with CALR, FAM134B, SEC62 and HSPB1, and the engagement of the autophagy core machinery through LC3B (selected papers are: *J Biol Chem* 2012; 287, 29921-29930. PMID: PMC3436137; *Aging Cell* 2018; 17: e12820. PMID: PMC6156544; *iScience* 2021; 24; 102315. PMID: PMC8042170; and *Commun Biol* 2021; 4: 454. PMID: PMC8041774).

These functions were dissected in mouse models of reduced (AT-1^{S113R/+}) and increased (AT-1 nTg and AT-1 sTg) ER acetylation (see: *J Exp Med* 2016; 213: 1267-1284. PMID: PMC4925020; *J Neurosci* 2014; 34: 6772-6789. PMID: PMC4019794; *Aging Cell* 2018; 17: e12820. PMID: PMC6156544) as well as in *Atase1*^{-/-} and *Atase2*^{-/-} mice (see *Commun Biol* 2021; 4: 454. PMID: PMC8041774). The specific roles of ATase1 and ATase2 in the regulation of reticulophagy is reported in *Commun Biol* 2021; 4: 454. PMID: PMC8041774 while the ability of ATase inhibitors to stimulate reticulophagy is reported in *Aging Cell* 2018; 17: e12820. PMID: PMC6156544.

Figure 6 of the present manuscript provides target engagement information that builds from the above already published data/papers.

Or perhaps another reticulophagy-inducing drugs other than ATase inhibitors (if there are any) might be used to mimic rescue phenotypes of AT-1 sTg mice.

Response: As far as we know, ATase inhibitors are the only reticulophagy-specific inducing drugs.

Minor comments:

1. *In previous paper (Peng et al., Aging Cell 2018), the authors stated that Compound 9 was administered at weaning when “the initial disease phenotypes were already manifested.” This seems to contradict the claim made in this paper (lines 128-130), which states that the previous*

paper showed that Compound 9 prevented the development of disease phenotypes. This portion should be revised or clarified.

Response: This has been clarified.

2. Lines 170-171 state that acetyl-CoA is one of the natural ATase substrates. What are other natural substrates of ATase and does Compound 9 block binding of ATase to those substrates as well?

Response: As far as we know, acetyl-CoA is the only natural substrate of the ATases. The sentence has been corrected. We thank the Reviewer for noticing our error.

3. Lines 176-177 are a little unclear. Expand upon how the commonalities and differences can illuminate their biochemical properties.

Response: We agree with the Reviewer. The text has been modified accordingly.

4. By what reasoning did the authors chose to further explore Compounds 10, 11, and 19? The basis for choosing those three seems to be on binding affinity assessment (line 168), but the results of that for the other compounds are not shown. It would be helpful to see the results for the rest of the compounds in the supplementary material.

Response: The Reviewer is correct. We selected Compounds 10, 11 and 19 based on their binding affinities and their druggability properties. The data for Compounds 9, 10, 11, and 19 are already included in Figure 3. Additional properties (as well as the results for all other compounds) have been included in the Supplementary Material (see Supplementary Tables 1-2).

5. Line 177. Cite Lipinski's Rule.

Response: Done as requested.

6. Supplementary Figures should be properly labeled.

Response: They are already labeled at the bottom of the page.

Reviewer #2

1) Is there any difference in the housing/handling of the AT-1 sg control animals compared to the treatment groups in Fig.5b prior to the onset of treatment? Particularly in the females it appears there is divergence in survival prior to commencing dosing with the drugs.

Response: There is no difference in the housing/handling of the females. The lifespan of males and females AT-1 sTg is essentially similar (see Fig. 1b, red line; Fig. 4b, blue line; see also our *Aging Cell* 2018 paper)

2) Several of the the lead compounds, particularly C9 share many features with previously described neuronal autophagy-stimulating compounds (Svetkov et al., *PNAS* 2010 PMID:20833817). Do the most potent compounds in this series (10-NCP and trifluoperazine) show *in silico* binding activity for ATase 1/2 and if so how do they compare to C9/C19?

Response: They do not display significant *in silico* binding for ATase1 or ATase2.

3) Please provide survival data for the entire cohort used in Fig.7 to show whether C19's increased specificity for ATase1 is therapeutically beneficial.

Response: Figure 6 (previously, Fig. 7) only shows that “target engagement” of Compound 9 and 19 overlaps with the phenotypic rescue. These animals were sacrificed at the indicated time-points to perform the analysis shown in Fig. 6b-c and 6e-f. The lifespan of the cohorts used to determine therapeutic efficacy of the different compounds are shown elsewhere (see Fig. 1b and Fig. 4b).

Reviewer #3

*The only experiment that I would suggest that would improve the impact of the work is testing levels of AT-1 in young and aged mice to complement the *in vitro* fibroblast analysis.*

Response: Done as requested.

All other comments are minor.

Fig 1b the red dots should be moved in front of green so that both can be observed by reader.

Response: The green lines begin at treatment (2 months of age). The initial cohort of 86 mice (for males) and 90 mice (for females) was randomly split in two cohorts, one untreated (43 males and 45 females) and one treated (43 males and 45 females).

Line 186 where the authors state all 3 compounds rescue phenotype does not agree with the information presented in fig with compound 11. Compound 11 doesn't appear to rescue - perhaps authors should say initial observations suggest all 3 rescue but with evaluation of

additional animals compound 11 has variable rescue level. This is stated later but would be more consistent with fig if clarified early in paragraph.

Response: We do say that Compound 11 was less effective and showed variability in the degree of rescue across all parameters. However, we must point out that the lifespan (see Fig. 4b, green line) and the other outcomes (see remaining panels of Fig. 4) do show partial rescue.

Fig 2 please define which 2 experimental groups are being compared for statically significance. Presumably the compound treated mice are compared to untreated and show statistically significant difference.

Response: The Reviewer is correct. We added the sentence “*Black symbol, significance vs WT; Red symbol, significance vs untreated AT-1 sTg*” in the legend of relevant figure.

REVIEWERS' COMMENTS:

Reviewer #1 (Remarks to the Author):

The authors made noticeable improvements in their manuscript through revision.

Reviewer #2 (Remarks to the Author):

The authors have adequately addressed point #2 and #3.

For concern #1 they have conflated two separate queries. A) survival differences between the AT-1sg control and treatment animals prior to the onset of treatment and B) sex differences in survival.

I am satisfied with the response to part B).

For Part A) it remains unclear why there is survival difference for the AT-1sg group (as a whole - both sexes together) prior to the onset of treatment. Were these groups treated identically prior to treatment? Either provided appropriate statistical analysis showing that there is no difference or comment on this unexpected data in the result/discussion.